# Design of the Crosslinking Reactions for Nucleic Acids-Binding Protein and Evaluation of the Reactivity

**Kenta Odaira, Ken Yamada, Shogo Ishiyama, Hidenori Okamura and Fumi Nagatsugi ***

Institute of Multidisciplinary Research for Advanced Materials, Tohoku University, Sendai 980-8577, Japan; k.odaira@kobayashi.co.jp (K.O.); Ken.Yamada@umassmed.edu (K.Y.); rare.earth.144@gmail.com (S.I.); hidenori.okamura.b8@tohoku.ac.jp (H.O.)

**\*** Correspondence: nagatugi@tohoku.ac.jp; Tel.: +81-22-217-5633

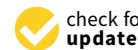

**Featured Application: Alkylation for nucleic acids-binding protein.**

**Abstract:** Selective chemical reactions of biomolecules are some of the important tools for investigations by biological studies. We have developed the selective crosslinking reactions to form covalent bonds to DNA or RNA using crosslinking oligonucleotides (CFO) bearing reactive bases. In this study, we designed the cross-linkable 4-amino-6-oxo-2-vinyltriazine derivative with an acyclic linker (*acy*AOVT) to react with the nucleic acids-binding protein based on our previous results. We hypothesized that the *acy*AOVT base would form a stable base pair with guanine by three hydrogen bonds at the positions of the vinyl group in the duplex DNA major groove, and the vinyl group can react with the nucleophilic species in the proximity, for example, the cysteine or lysine residue in the nucleic acids-binding protein. The synthesized oligonucleotides bearing the *acy*AOVT derivative showed a higher reactivity than that of the corresponding pyrimidine derivative without one nitrogen. The duplex containing *acy*AOVT-guanine (G) formed complexes with Hha1 DNMT even in the presence of 2-mercaptoethanol. We expect that our system will provide a useful tool for the molecular study of nucleic acids-binding proteins.

**Keywords:** oligonucleotide; crosslink; nucleic acid binding protein; cytosine methyltransferase

## 1. Introduction

The binding of proteins to nucleic acids (DNA and RNA) is central to all aspects of gene expression regulation. DNA-binding proteins include the transcription factor to bind to specific DNA sequences and control the transcription rate of genetic information from the DNA to RNA [1,2]. Many DNA modifying enzymes are also included in the DNA binding protein, for example, repair enzymes [3,4] and epigenetic modification enzymes. DNA methyl transferase is one of the epigenetic modification enzymes and catalyses the transfer of the methyl group to DNA modifying the function of genes and affecting gene expression [5,6]. RNA-binding proteins have important functions in the post-transcriptional process, such as splicing regulation [7] and modulation of the mRNA translation. Recent studies have revealed that post-transcriptional gene regulation by non-coding RNAs is involved in most biological activities. The association of the RNA binding protein with ncRNA plays a crucial role in these biological functions [8–10]. The chemical tools for the control of the interaction between the nucleic acids-binding protein and nucleic acids have the potential for the development as the new strategy for artificial control of the gene expression.

Crosslinking reactions between nucleic acids and binding proteins are powerful tools for analyzing these interactions. Photoreactive functional groups, such as diazirines and benzophenones,

were utilized for DNA or RNA-protein crosslinking [11–14], but lack selectivity for the target amino acid. Furthermore, due to their intrinsic reactivity, these groups react with water and other reactive chemical species resulting in low yields. Oligodeoxynucleotides (ODN) with reactive groups are exploited for their effective DNA-protein crosslinking reactions by the proximity effect of the specific DNA recognition with the binding protein. The reactive groups activated by oxidation, such as a diol [15] and furan [16], react with a proximal lysine or arginine in the protein or peptides. Hocek et al. reported that the ODN bearing a vinylsulfone amide [17] or chloroacetamide group [18] efficiently cross-linked with the p53 protein through alkylation of the cysteine in the proximal position with the reactive groups. Recently, they demonstrated that ODN with 2-vinylhypoxantine reacted with a thiol-containing minor groove binding peptide by proximity effect [19].

In our previous study, we reported the synthesis of the ODN bearing 4-amino-6-oxo-2-vinyltriazine with an ethyl linker (Et-AOVT: (**1**)) as a cross linkable derivative and an evaluation of its properties [20]. ODN showed a relatively high reactivity with the pyrimidine nucleobases of the complementary DNAs and the lowest reactivity to guanine. We hypothesized that the AOVT base would form a stable base pair with guanine (G) by three hydrogen bonds to be positioned at the vinyl group in the DNA major groove. In this study, we have designed an AOVT derivative with an acyclic linker (*acy*AOVT: (**2**)) as a cross-linkable probe of the nucleic acids-binding protein. The distance between the reactive base and sugar moiety of **2** is shorter than that of **1**, and consequently, the *acy*AOVT (**2**) should form a base pair with guanine (G) similar to the natural cytosine (C)-G pair. This assumption is supported by the molecular modeling shown in Figure 1. The molecular modeling revealed that the structure of the base pair of *acy*AOVT (**2**)-G and the natural C-G significantly overlap each other (Figure 1C). On the other hand, the structure of Et-AOVT (**1**)-G is subtly shifted from the natural base pair one (Figure 1B). The duplex DNA containing the *acy*AOVT (**2**)-G is expected to bind to the nucleic acid binding proteins similar to the natural duplex and form covalent linkages with the nucleophilic residues of the protein in the proximal position of the vinyl group (Figure 2).

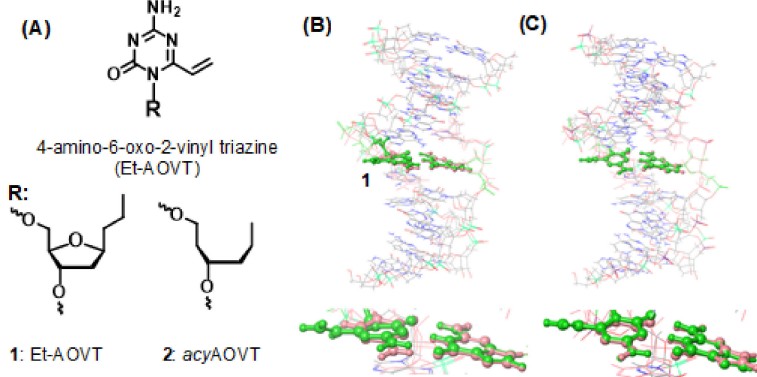

**Figure 1.** (**A**) Structures of cross-linkable derivatives; (**B**,**C**) Superimposed structures between the natural duplex (pink) and the duplex contained **1** (**B**) or **2** (**C**) (green).

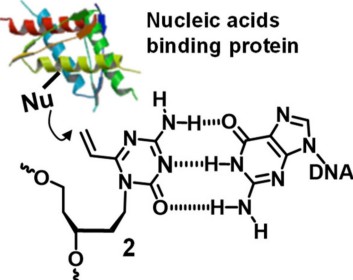

**Figure 2.** Design of the cross-linkable probe to nucleic acids-binding protein.

## 2. Materials and Methods

### 2.1. General

The $^1$H-NMR spectra were obtained by 400 or 600 MHz spectrometers (Bruker, Billerica, MA, USA). The $^1$H chemical shifts are described as $\delta$ values in ppm relative to acetone-d$_6$ (2.05 ppm), DMSO-d$_6$ (2.50 ppm), CDCl$_3$ (7.26 ppm), and tetramethylsilane (1H, 0.00 ppm). The $^{13}$C NMR spectra were obtained by a Bruker 600 MHz spectrometer. The $^{13}$C chemical shifts are described as the $\delta$ values in ppm relative to acetone-d$_6$ (29.84 ppm), DMSO-d$_6$ (39.52 ppm), and CDCl$_3$ (77.16 ppm). The $^{31}$P-NMR spectra were recorded by a Bruker 500 MHz (202 MHz for $^{31}$P). The multiplicity and qualifier abbreviations are as follows: s = singlet, d = doublet, t = triplet, q = quartet, quin. = quintet, sept. = septet, m = multiplet, br = broad. The electrospray ionization (ESI) mass spectra were recorded using a BioTOF II mass spectrometer or APEX III (Bruker Daltonics, Bruker, Billerica, MA, USA). The matrix-assisted laser desorption/ionization (MALDI-TOF) mass spectra were recorded Autoflex speed mass spectrometer and a laser at 337 nm in the negative mode using 3-hydroxypicolinic acid as the matrix or in the positive mode using 2,5-dihydroxybenzoic acid as the matrix. Thin-layer chromatography (TLC) was performed using silica gel 60 F$_{254}$ pre-coated plates. Column chromatography was performed using silica gel 60 N (spherical, neutral, 100–210 μm, Kanto Chemical, Tokyo, Japan). Flash chromatography was performed using Kanto Chemical silica gel 60 N (spherical, neutral, 40–50 μm). The ultraviolet-visible (UV-vis) absorption spectra were recorded using a DU800 spectrometer (Beckman Coulter, Brea, CA, USA). The ODN synthesis was carried out using an automated DNA synthesizer (392 DNA/RNA synthesizer, ABI, Foster City, CA, USA) following the standard phosphoramidite chemistry. High performance liquid chromatography (HPLC) was performed using a Cosmosil 5C18MSII column (4.6 or 10 × 250 mm, Nacalai Tesque, Kyoto, Japan), a PU-986 pump (JASCO, Tokyo, Japan), JASCO 2075 UV detector and a JASCO 2067 column oven. pH measurements were measured by a Seven Easy pH meter (Mettler Toledo, Columbus, OH) using an 8220BNWP electrode. The denaturing polyacrylamide gel plates were visualized and quantified using a FLA-5100 Fluor Imager (Fujifilm, Tokyo, Japan). Anhydrous methanol, DMF, THF, CH$_2$Cl$_2$, dioxane, pyridine, DMSO, CH$_3$CN, toluene and THF were purchased from Wako Pure Chemical Industries, Ltd. (Tokyo, Japan). Unless otherwise noted, all the synthetic reactions were carried out at ambient temperature. The reactions requiring anhydrous conditions were achieved under an argon atmosphere in flasks dried under 1–3 mmHg. Commercially available reagents were obtained from Wako Pure Chemical Industries, Ltd., TCI, Inc., (Tokyo, Japan), Sigma-Aldrich Co. LLC (St. Louis, MO) and Kanto Chemical Co., Inc (Tokyo, Japan) and used without further purification. Human DNA (cytosine-5) methyltransferase (DNMT1) was obtained from New England Bio Labs Japan, Inc. (Tokyo, Japan)

### 2.2. Synthesis of the Nucleoside Derivatives

#### 2.2.1. Synthesis of 4-amino-6-oxo-1-(3′,5′-*O*-t-butyldimethylsilyl-2′,4′-dideoxy-ᴅ-ribityl) triazine (**4**)

The linker **3** (1.3 g, 3.0 mmol), which was prepared by a modified literature procedure [21], was added to 5-aza cytosine Na salt (1.0 g, 7.5 mmol) in DMSO (30 mL). The mixture was stirred at 60 °C for 18 h. The resulting mixture was diluted with EtOAc and quenched with NH$_4$Cl. The product was extracted with EtOAc and the combined organic layer was washed with brine, dried over Na$_2$SO$_4$, filtered, and evaporated under reduced pressure. The resulting oil was purified by silica gel column chromatography (CH$_3$OH/CHCl$_3$, 1:100, then CH$_3$OH/CHCl$_3$, 1:50) to afford the corresponding N1-alkylation (772 mg, 1.74 mmol, 58%) and N3-alkylation (126 mg, 0.29 mmol, 10%) products.

N1-alkylation product: $^1$H-NMR (400 MHz, CDCl$_3$) $\delta$ 7.90 (s, 1H), 6.98 (brs, 1H), 5.65 (brs, 1H), 3.97 (ddd, 2H, J = 12.0, 12.0, 6.0 Hz), 3.90–3.77 (m, 2H), 3.64 (dd, 2H, J = 6.4, 6.4 Hz), 1.99–1.60 (m, 6H), 1.37–1.23 (m, 1 H), 0.89 (s, 9H), 0.87 (s, 9H), 0.07 (s, 3H), 0.06 (s, 3H), 0.03 (s, 6H); $^{13}$C-NMR (125 MHz, CDCl3) $\delta$ 166.7, 158.7, 154.5, 67.4, 59.6, 45.1, 40.2, 36.2, 26.12, 26.09, 18.4, 18.3, −4.2, −4.3, −5.14, −5.16; HRMS (ESI) calcd. for C$_{20}$H$_{43}$N$_4$O$_3$Si$_2$$^+$ [M + H]$^+$ *m/z* 443.2868, found *m/z* 443.2879.

N3-alkylation product: $^1$H-NMR (600 MHz, CDCl3) δ 8.34 (s, 1H), 5.90 (brs, 1H), 5.61 (brs, 1H), 4.42–4.35 (m, 1H), 4.05–4.01 (m, 1H), 3.70–3.64 (m, 2H), 1.99–1.95 (m, 1H), 1.89–1.84 (m, 2H), 1.7–1.67 (m, 2H), 0.871 (s, 9H), 0.867 (s, 9H), 0.046 (s, 3H), 0.027 (s, 3H), 0.025 (s, 3H), 0.018 (s, 3H); $^{13}$C-NMR (125 MHz, CDCl3) δ 170.4, 167.93, 167.88, 66.3, 64.5, 59.6, 40.3, 35.9, 25.9, 25.8, 18.2, 18.0, −4.6, −4.8, −5.36, −5.38; HRMS (ESI) calcd. for $C_{20}H_{43}N_4O_3Si_2{}^+$ [M + H]$^+$ *m/z* 443.2868, found *m/z* 443.2879

### 2.2.2. Synthesis of 4-amino-2-(2-octylthioethyl)-6-oxo-1-(3′,5′-*O*-t-butyldimethylsilyl-2′, 4′-dideoxy-ᴅ-ribityl) triazine (**6**)

A 28% aq NH$_4$OH (10 mL) solution was added to a solution of the N1-alkylation product **4** (172 mg, 390 μmol) in dioxane-methanol (10 mL, 1/1, *v/v*), then sealed with a glass stopper. The mixture was stirred at 55 °C for 24 h. The resulting solution was evaporated under vacuum at 43 °C. The resulting moist solid was co-evaporated with ethanol × 3 to afford guanylurea **5**. This product was directly used for the next step without further purification. The orthoester (597 mg, 1.95 mmol, purity = 91%) was added to a solution of guanylurea (169 mg, 390 μmol) in DMF (0.2 M, 1.95 mL). The mixture was stirred at 120 °C for 2 h. The resulting yellow solution was diluted with Et$_2$O. The product was extracted with Et$_2$O and the combined organic layer was washed with NH$_4$Cl aq., NaHCO$_3$ aq., and brine, dried over Na$_2$SO$_4$, filtered and evaporated. The crude product was purified by silica gel column chromatography (CHCl$_3$ then, CHCl$_3$ with 1% Et$_3$N, then CH$_3$OH/CHCl$_3$, 1:300 with 1% Et$_3$N) to afford **6** (198.3 μmol, 51%) as a yellow oil: $^1$H-NMR (600 MHz, CDCl$_3$) δ 5.70 (brs, 1H), 5.17 (brs, 1H), 4.02–3.89 (m, 3H), 3.65 (dd, 2H, *J* = 6.6, 6.6 Hz), 2.95–2.88 (m, 3H), 2.54 (t, 2H, *J* = 7.8 Hz), 1.93–1.88 (m, 1 H), 1.79–1.64 (m, 4H), 1.59 (quin, 2H, *J* = 7.2 Hz), 1.39–1.25 (m, 10H), 0.91 (s, 9H), 0.89 (t, 3H, *J* = 7.2 Hz), 0.88 (s, 9H), 0.089 (s, 3H), 0.086 (s, 3H), 0.04 (s, 6H); $^{13}$C-NMR (150 MHz, CDCl$_3$) δ 168.8, 165.3, 155.8, 67.6, 59.6, 41.6, 40.4, 35.8, 34.3, 33.0, 32.1, 29.9, 29.5, 29.2, 28.2, 26.19, 26.18, 22.9, 18.5, 18.3, 14.4, 0.26, −4.18, −4.26, −4.30, −5.1; HRMS (ESI) calcd. for $C_{30}H_{63}N_4O_3SSi_2{}^+$ [M + H]$^+$ *m/z* 615.4154, found *m/z* 615.4158.

### 2.2.3. Synthesis of 2-(2-octylthioethyl)-6-oxo-4-acetylamino-1-(3′,5′-*O*-t-butyldimethylsilyl-2′, 4′-dideoxy-ᴅ-ribityl) triazine (**7**)

AcCl (5.3 μL, 74.7 μmol) was added to a solution of **6** (15 mg, 24.9 μmol) in pyridine (250 μL). The mixture was stirred at rt for 19 h. The resulting solution was diluted with Et$_2$O and quenched with NH$_4$Cl. The product was extracted with Et$_2$O and the combined organic layer was washed with brine, dried over Na$_2$SO$_4$, filtered and evaporated. The crude product was purified by silica gel column chromatography (CHCl$_3$, then CHCl$_3$ with 1% Et$_3$N) to afford **7** (13.4 μmol, 54%) as a colorless oil: $^1$H-NMR (600 MHz, CDCl$_3$) δ 7.71 (br, 1H), 4.08–3.98 (m, 3H), 3.66 (t, 2H, *J* = 6.0 Hz), 3.65 (t, 2H, *J* = 6.0 Hz), 3.00 (t, 2H, *J* = 6.6 Hz), 2.91 (t, 2H, *J* = 6.6 Hz), 2.61 (s, 3H), 2.54 (t, 2H, *J* = 7.2 Hz), 1.95 (m, 1H), 1.81–1.64 (m, 3H), 1.58 (qt, 2H, *J* = 7.2, 7.2 Hz), 1.39–1.26 (m, 10 H), 0.92 (s, 9H), 0.91 (t, 3H, *J* = 7.2 Hz), 0.88 (s, 9H), 0.095 (s, 3H), 0.093 (s, 3H), 0.041 (s, 3H), 0.03 (s, 3H); $^{13}$C-NMR (150 MHz, CDCl$_3$) δ 172.0, 170.2, 161.5, 154.8, 41.9, 40.0, 34.9, 34.1, 32.8, 29.7, 29.5, 29.2, 28.8, 27.7, 25.87, 25.86, 25.85, 22.6, 18.2, 18.0, 14.1, −4.52, −4.55, −5.38, −5.41; HRMS (ESI) calcd. for $C_{32}H_{64}N_4O_4SSi_2{}^+$ [M + H]$^+$ *m/z* 657.4246, found *m/z* 657.4246.

### 2.2.4. Synthesis of 2-(2-octylthioethyl)-6-oxo-4-acetylamino-1-(2′,4′-dideoxy-ᴅ-ribityl) triazine (**8**)

Boron trifluoride-diethylether complex (51 μL, 398 μmol) was slowly added to a solution of **7** (131 mg, 199 μmol) in CH$_3$CN (800 μL) at 0 °C. After the addition, the mixture was stirred at 0 °C for 2 h. The resulting solution was diluted with EtOAc, then quenched with phosphate buffer (pH = 6.0, 1 M). The product was extracted with EtOAc and the combined organic layer was washed with brine, dried over Na$_2$SO$_4$, filtered and evaporated. The crude product was purified by silica gel chromatography, eluting with CH$_3$OH/CHCl$_3$, 1:50, then 1:10 to give **8** (103 μmol, 52%) as a white solid: $^1$H-NMR (600 MHz, CDCl$_3$) δ 7.80 (br, 1H), 4.38–4.33 (m, 1H), 4.00–3.96 (m, 1H), 3.91–3.88 (m, 1H), 3.85–3.81 (m, 2H), 3.15 (dt, *J* = 16.8, 1H, 7.2 Hz), 3.05 (dt, 1H, *J* = 16.8, 7.2 Hz), 2.93 (t, 2H, *J* = 7.2 Hz), 2.61 (s, 3H), 2.56 (t, 2H, *J* = 7.2 Hz), 1.95–1.90 (m, 1H), 1.82–1.71 (m, 3H), 1.59 (qt, 2H, *J* = 7.2, 7.2 Hz), 1.38–1.25 (m, 10H),

0.88 (t, 3H, *J* = 7.2 Hz); $^{13}$C-NMR (150 MHz, CDCl$_3$) δ 171.9, 170.9, 161.7, 156.0, 68.1, 61.63, 61.59, 41.7, 38.0, 36.2, 34.1, 32.8, 31.8, 29.6, 29.2, 28.8, 27.9, 25.9, 22.6,14.1; HRMS (ESI) calcd. for C$_{20}$H$_{36}$N$_4$O$_4$S$^+$ [M + H]$^+$ *m/z* 429.2528, found *m/z* 429.2530.

### 2.2.5. Synthesis of 2-(2-octylthioethyl)-6-oxo-4-acetylamino-1-(5′-*O*-(4,4′-dimethoxytrityl)-2′, 4′-dideoxy-D-ribityl) triazine (**9**)

DMTrCl (58 mg, 170 μmol) was added to a solution of **8** (497 mg, 114 μmol) in pyridine (487 μL) at 0 °C. After the addition, the mixture was stirred at 0 °C for 1 h. The resulting mixture was diluted with CH$_2$Cl$_2$, then quenched with NaHCO$_3$. The product was extracted with CH$_2$Cl$_2$ and the combined organic layer was washed with brine, dried over Na$_2$SO$_4$, filtered and evaporated. The crude product was purified by chromatography on silica gel, eluting with CHCl$_3$ and 1% Et$_3$N, then CH$_3$OH/CHCl$_3$, 1:200 with 1% Et$_3$N to give **9** (80 μmol, 70%) as a colorless oil: $^1$H-NMR (600 MHz, CDCl$_3$) δ 7.72 (br, 1H), 7.39–7.38 (m, 2H), 7.30–7.27 (m, 6H), 7.21–7.19 (m, 1H), 6.83–6.80 (m, 2H), 4.20 (dt, 1H, *J* = 13.2, 7.2 Hz), 4.04 (dt, 1H, *J* = 13.2, 7.2 Hz), 3.15 (dt, 1H, *J* = 16.8, 7.2 Hz), 3.06 (dt, 1H, *J* = 16.8, 7.2 Hz), 2.96 (t, 2H, *J* = 7.2 Hz), 2.60 (s, 3H), 2.54 (t, 2H, *J* = 7.2 Hz), 1.96–1.91 (m, 1H), 1.83–1.77 (m, 1H), 1.69–1.66 (m, 2H), 1.57 (qt, 2H, *J* = 7.2, 7.2 Hz), 1.37–1.26 (m, 10H), 0.87 (t, 3H, *J* = 7.2 Hz); $^{13}$C-NMR (150 MHz, CDCl$_3$) δ 172.0, 170.8, 161.6, 158.5, 155.3, 144.5, 135.7 135.6, 129.88, 129.86, 128.0, 127.9, 126.9, 113.2, 86.9, 68.4, 62.4, 55.2, 42.1, 36.5, 35.5, 34.1, 32.6, 31.8, 29.5, 29.19, 29.17, 28.8, 25.9, 22.6, 14.1; HRMS (ESI) calcd. for C$_{41}$H$_{54}$N$_4$O$_6$S$^+$ [M + H]$^+$ *m/z* 731.3837, found *m/z* 731.3834.

### 2.2.6. Synthesis of 2-(2-octylthioethyl)-6-oxo-4-acetylamino-1-(3′-*N*,*N*-diisopropyl cyanoethyl-phosphoramidyl-5′-*O*-(4,4′-dimethoxytrityl)-2′,4′-dideoxy-D-ribityl) triazine (**10**)

DIPEA (83.2 μL, 14.3 μmol) was added to a solution of **9** (15.2 mg, 20.8 μmol) in CH$_2$Cl$_2$ (416 μL) and the mixture was cooled to 0 °C. NCCH$_2$CH$_2$OP[(N(*i*-Pr)$_2$]Cl (9.3 μL, 41.6 μmol) was then added to the mixture. After the addition, the mixture was stirred at 0 °C for 1 h, diluted with CH$_2$Cl$_2$, then quenched with NaHCO$_3$. The product was extracted with CH$_2$Cl$_2$ and the combined organic layer was washed with brine, dried over Na$_2$SO$_4$, filtered and evaporated. The crude product was co-evaporated with toluene, then purified by chromatography on silica gel, eluting with EtOAc/hexane, 1:1 with 1% Et$_3$N, then 2:3 with 1% Et$_3$N to give **10** (15.0 μmol, 72%) as a colorless oil mixture of two diastereomers: $^1$H-NMR (600 MHz, CDCl$_3$) δ 7.65 (br, 1H), 7.41 (d, 2H, *J* = 7.2 Hz), 7.31–7.27 (m, 6H), 7.21–7.18 (m, 1H), 6.83–6.81 (m, 4H), 4.22–4.04 (m, 3H), 3.82–3.49 (m, 4H), 3.79 (s, 6H), 3.22–3.15 (m, 2H), 3.10–2.95 (m, 2H), 2.91–2.86 (m, 2H), 2.64–2.61 (m, 1H), 2.62–2.61 (m, 3H), 2.55–2.50 (m, 2H), 2.48–2.46 (m, 1H), 2.01–1.77 (m, 4H), 1.59–1.54 (m, 2H), 1.39–1.25 (m, 10H), 1.16–1.03 (m, 12H), 0.88 (t, 3H, *J* = 7.2 Hz); $^{13}$C-NMR (150 MHz, CDCl$_3$) δ 172.04, 172.00, 170.4, 170.2, 161.44, 161.40, 158.3, 154.9, 154.7, 145.10, 145.05, 136.32, 136.30, 136.29, 136.27, 130.0, 128.1, 128.0, 127.7, 126.7, 126.6, 118.0, 117.6, 113.0, 86.0, 77.2, 77.0, 76.8, 70.4, 70.3, 70.0, 69.9, 60.4, 60.0, 58.0, 57.8, 57.4, 57.3, 55.18, 55.15, 43.08, 43.05, 43.00, 42.97, 42.0, 41.9, 36.82, 36.80, 36.25, 36.23, 34.4, 34.3, 34.2, 34.04, 33.95, 32.82, 32.79, 31.8, 29.53, 29.52, 29.18, 29.16, 28.84, 28.83, 27.7, 25.9, 24.80, 24.75, 24.60, 24.56, 24.52, 24.50, 22.6, 21.1, 20.50, 20.45; $^{31}$P- NMR (202 MHz, CDCl$_3$) δ 147.70, 146.72; HRMS (ESI) calcd. for C$_{50}$H$_{71}$N$_6$O$_7$PS$^+$ [M + H]$^+$ *m/z* 931.4915, found *m/z* 931.4919.

### 2.3. Synthesis of the Oligodeoxynucleotides Containing acyAOVT Derivatives

**ODN1** and **2** were synthesized on a 1 μmol scale by an ABI 392 DNA/RNA synthesizer with standard β-cyanoethyl chemistry. 5′-Terminal dimethoxytrityl-bearing ODN**1** and **2** was removed from the solid support by treatment with 45 mM K$_2$CO$_3$-MeOH containing 10 mM 1-octanethiol (0.5 mL) and the residue was evaporated under reduced pressure. The crude product was purified by reverse phase HPLC with a C-18 column (Nacalai Tesque: Cosmosil 5C18-MS-II, 10 × 250 mm) by a linear gradient of 5–40%/25 min of acetonitrile in 0.1% TEAA buffer at the flow rate of 4 mL/min. The dimethoxytrityl group of the purified **ODN** was removed with 10% AcOH for 30 min and the mixture was additionally purified by reverse phase HPLC to afford **ODN1**; MALDI-TOF MS (*m/z*):

calcd. for [M-H]⁻ *m/z* 4045.8746; found 4045.780; **ODN2**; MALDI-TOF MS (*m/z*): calcd. for [M-H]⁻ *m/z* 6960.8115; found 6958.589.

To a solution of **ODN1** or **2** (70 μM) in ddH$_2$O was added a solution of MMPP (1 mM) in carbonate buffer (pH = 10) at room temperature. After 1 h, 5% AcOH was added to the mixture and the mixture was left for an additional 2 h to give **ODN3**. MALDI-TOF MS ODN (SOOct): calcd. for [M-H]⁻ *m/z* 4061.8736, found 4061.755, **ODN5** (vinyl): calcd. for [M-H]⁻ *m/z* 3899.5826, found 3899.304; **ODN6** (vinyl): calcd. for [M-H]⁻ *m/z* 6814.5195, found 6811.219

ODN**7** was prepared by using 1.0 μM ODN**5** and 100 mM sodium thiomethoxide in 50 mM MES buffer (pH = 7.0) at 37 °C for 19 h. The reaction mixture was purified by reverse phase HPLC to afford ODN**7**: calcd. for [M-H]⁻ *m/z* 3947.6856, found 3947.309.

### 2.4. General Procedure for the Crosslinking Reactions

The reaction was performed using 5.0 μM **ODN5** and 2.0 μM of the target DNA or RNA labelled by fluorescein at the 5′-end in a buffer of 100 mM NaCl and 50 mM MES buffer (pH = 7.0). The reaction was incubated at 37 °C for 1–24 h. The reaction was quenched with the addition of loading dye (95% formamide, 20 mM EDTA, 0.05% xylene cyanol, and 0.05% bromophenol blue). The cross-linked products were analyzed by a denaturing 20% polyacrylamide gel electrophoresis containing urea (7 M) with TBE buffer at 200 V for 1 h. The labelled bands were visualized and quantified using a FLA-5100 Fluor Imager.

### 2.5. $T_m$ Measurement

All samples for the $T_m$ measurements consisted of 100 mM NaCl, 25 mM MES buffer (pH = 7.0) and 1 μM Duplex. The $T_m$ measurements were performed using a temperature controller. Both the heating and cooling curves were measured three times over the temperature range of 25 °C to 80 °C at 0.5 °C/min. The absorbance at 260 nm was recorded every 0.5 °C.

### 2.6. Alkylation with ODN Probe to HhaI DNMT-1

The duplex oligo (final 1 μM) in pH 7.5 reaction buffer (50 mM Tris-HCl, 10 mM EDTA, 5 mM 2-mercaptoethanol) was pre-incubated at 37 °C for 5 min. HhaI Methyltransferase was then added to the reaction mixture and the mixture was pre-incubated at 37 °C (or 4 °C) for another 5 min. SAM was added to the mixture and the mixture was incubated at 37 °C (or 4 °C) for 0.5–24 h. A 4.5 μL aliquot of the reaction mixture (4.5 μL: for silver stain, 12 μL: for Flamingo gel stain) was used every time. Loading buffer (6×Tris.HCl, SDS, glycerol, 0.05% bromphenol blue, 200 mM DTT (final 20 mM)) was added and the mixture was heated at 65 °C for 15 min. The samples were separated by 10% SDS-PAGE (0.025 M Tris, 0.192 M glycine, 0.1% SDS) at room temperature (10 mA/1.5 h then 20 mA/1.0 h). Visualization of the 5′- FAM oligonucleotide was performed by fluorescence imaging using an FLA-5100 Fluor Imager. Visualization of the protein was performed by silver stain or Flamingo gel stain.

## 3. Results

### 3.1. Synthesis of the acyAOVT Nucleoside Phosphoramidite

We planned to synthesize the C6-substituted 5-azacytosine by ring opening of the 5-azacytosine moiety forming guanylurea and subsequent ring closing reaction that follows a previous report (Scheme 1) [20]. In our previous study, the acyclic linkers-protected MOM or Bn group produced the coupling products in low yields. The acyclic side chain **3**-protected TBS group was synthesized from 2′-deoxy-ᴅ-ribose by a previous modified procedure [21]. The coupling reaction between the sodium salt of 5-azacytosine and the acyclic side chain compound **3** was done to form the N1- and N3- alkylated products. The reaction conditions were screened in order to optimize the formation of the N1-alkylated product **4** and a considerable amount of the N1-alkylation (58%) in DMSO at 60 °C

was achieved with a N1:N3 ratio = 6:1. The formation of the glycosidic bond at the N1 position of 5-azacytosine was confirmed by HMQC and HMBC analyses.

**Scheme 1.** *Reagents and conditions*: (**a**) 5-azacytosine sodium salt, DMSO, 60 °C, 18 h, 58%; (**b**) 7M NH$_4$OH, MeOH-1,4-dioxane (1:1), 55 °C, 24 h; (**c**) orthoester, DMF, 120 °C, 2 h, 51% (2 steps); (**d**) AcCl, pyridine, rt, 19 h, 54%; (**e**) BF$_3$·OEt, CH$_2$Cl$_2$, 0 °C, 2 h, 52%; (**f**) DMTrCl, pyridine, 0 °C, 1 h, 70%; (**g**) 2-Cyanoethyl-*N,N*-diisopropylchloropho sphoramidite, DIPEA, CH$_2$Cl$_2$, 0 °C, 1 h, 72%.

The 5-azacytidine derivative **4** was treated with NH$_3$ to give the guanylurea intermediate **5**, which was condensed with the orthoester to afford the desired C6-octylthioethyl-5-azacytidine derivative **6** in 45% yield (2 steps). After the 4-*N*-acetylation and the deprotection of the TBS groups by BF$_3$OEt$_2$, the 5′-hydroxyl group was selectively protected with the DMTr group, then the 3′-hydroxyl group was phosphitylated to yield the *acy*AOVT nucleoside phosphoramidite **10**.

*3.2. Synthesis of the Oligonucleotides Containing acyAOVT and Evaluation of the Crosslinking Reactivity*

We synthesized two kinds of **ODNs1**, **2** by a DNA synthesizer using phosphoramidite **10** (Scheme 2). The synthesized ODNs were treated with 45 mM K$_2$CO$_3$/MeOH containing 10 mM 1-octanethiol to cleave them from the resin. After the DMTr-ON purification by reverse-phase (RP) HPLC, detritylation was carried out in an aqueous 10% AcOH solution at room temperature for 30 min. The obtained ODNs were further purified by RP-HPLC and were characterized by a MALDI-TOF MS analysis. The sulfide group of ODNs**1**, **2** was oxidized with magnesium monoperoxyphthalate (MMPP) and treated with aqueous 10% AcOH to give ODNs**5** and **6** according to the reported procedure [20]. The characterization of the ODNs was performed by MALDI-TOF MS.

**Scheme 2.** *Reagents and conditions*: (**a**) DNA synthesizer; (**b**) (i) 45 mM K$_2$CO$_3$-MeOH, 10 mM 1-octanethiol, rt, 4 h; (ii) 10% AcOH, rt, 30 min.; (**c**) 1 mM MMPP (2 eq) in carbonate buffer (pH 10), rt, 1 h; (**d**) 5% aq. AcOH, rt, 2 h.

The crosslinking reactions using the reactive ODN**5** to initially the complementary target DNA**1** or RNA**1** labelled with fluorescein at the 5′ end were investigated under neutral conditions. The reaction mixture was analyzed by 20% polyacrylamide gel electrophoresis (PAGE) containing 7 M urea and the yields of the cross-linked product were calculated based on the fluorescent intensity of each band observed on the gels. Figure 3 shows a comparison of the reactivity of ODN**5** toward the different bases at the target site of the DNA**1** or RNA**1**. ODN**5** showed relatively high crosslinking reactivity to

dC, dT, and dG, except for dA in the DNA. On the other hand, the crosslinked product was observed in high yields with rC and rU, and no significant products were observed with rG and rA in the RNA. The comparison of the reactivity to DNA or RNA with acyAOVT **2**, 4-amino-6-oxo-2-vinyl pyrimidine having an acyclic linker (**11:** acyAOVP) and Et-AOVT **1**, is shown in Figure 4. By comparison, acyAOVT **2** and acyAOVP **11** showed a significant difference in the reaction rate and the base selectivity to the target DNA and RNA (Figure 4A,B).

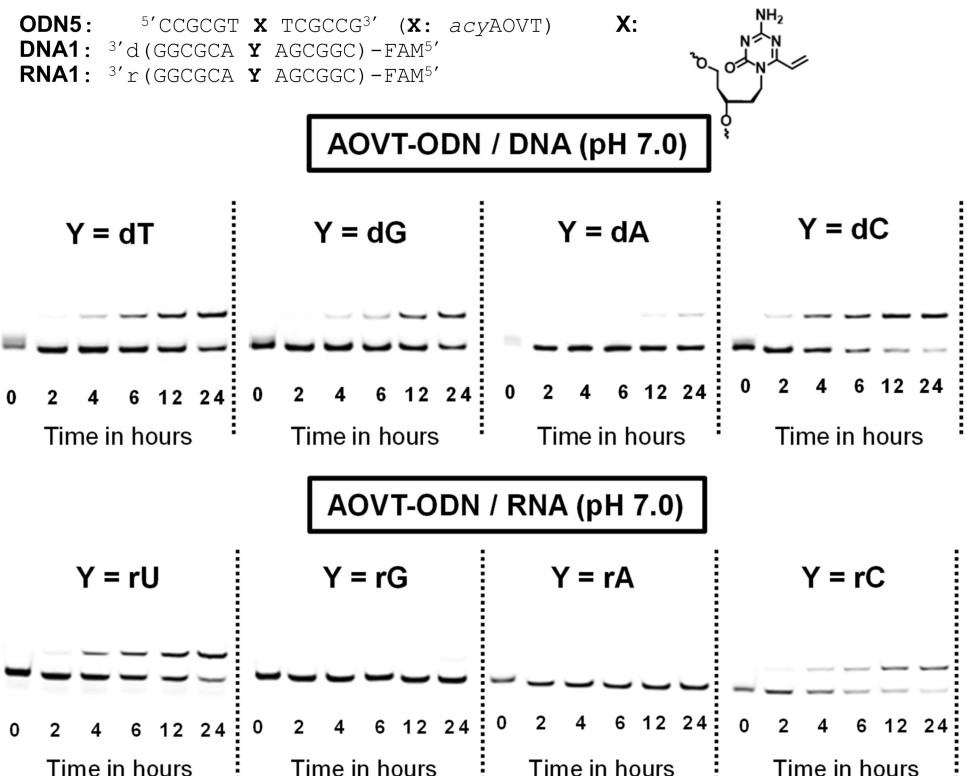

**Figure 3.** Denaturing gel electrophoresis of crosslink reaction to target DNA1 and RNA1 with **ODN5**. The reaction was performed in 50 mM MES-buffer (pH 7.0) containing 100 mM NaCl, **ODN5** (5 μM) and cDNAs or cRNAs (2 μM) at 37 °C.

The reaction rate with *acy*AOVT **2** was faster than that of *acy*AOVP **11**. The higher reaction rate with *acy*AOVT **2** might be attributed to the electron-withdrawing effect of the triazine increasing the reactivity of the 6-vinyl group of *acy*AOVT **2** than that of the pyrimidine-type *acy*AOVP **11**. Compared to the target base selectivity in **DNA1** with previously reported the *acy*AOVP **11** [21], AOVT **1** and **2** showed a drastically increased reactivity with the cytosine. On the other hand, the crosslinking yields to **RNA1** with AOVT **1** and **2** were significant higher to cytosine and uracil, and relatively lower to guanine than that of AOVP **11**. The reaction rate and selectivity to the pyrimidine bases (C, T and U) with *acy*AOVT **2** was comparable to that of Et-AOVT **1**. The reactivity to adenine (A) in DNA and RNA with **2** was significantly lower than that of Et-AOVT **1** (Figure 4A,C).

Next, we carried out a thermal denaturing study of the duplex containing the non-reactive *acy*AOVT. The **ODN7** bearing the non-reactive *acy*AOVT (SMe) was prepared by the addition of NaSMe to the vinyl derivative under weak acidic conditions for avoiding the decomposition of *acy*AOVT to guanylurea (Scheme 3).

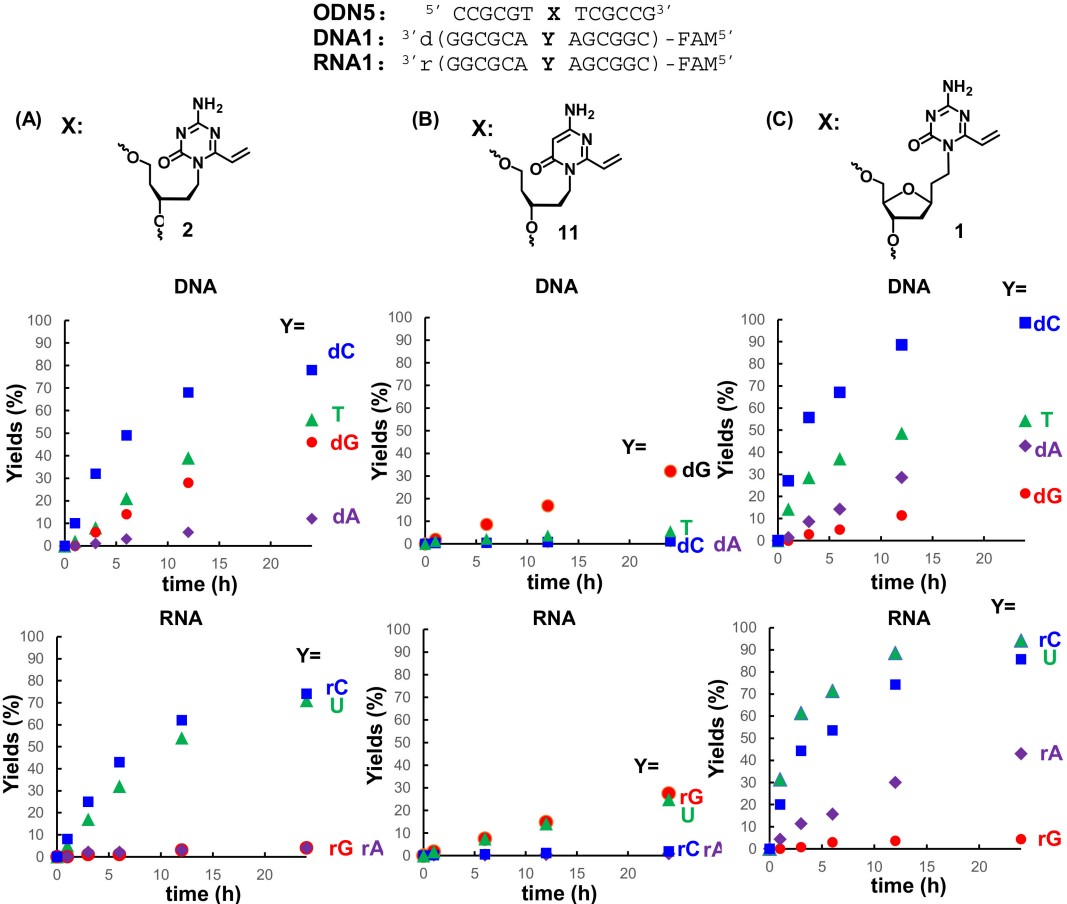

**Figure 4.** Comparison of the reaction yields calculated from the gel electrophoresis analysis of the cross-linking to target DNA**1** (**Y** = dT, dG, dC, dA) or RNA**1** (**N** = U, G, C, A) with *acy*AOVT (**2**) (**A**), *acy*AOVP (**11**) (**B**) and Et-AOVT (**1**) (**C**).

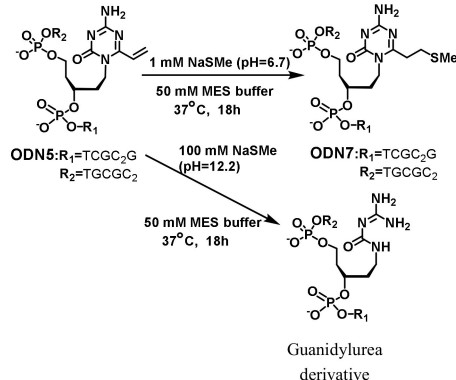

**Scheme 3.** Synthesis of the ODN bearing a stable precursor.

The melting temperature (*Tm*) of the **ODN7/DNA1** or **RNA1** is summarized in Figure 5. The *Tm* values of the **ODN7/DNA1** or **RNA1** were observed to be lower than that of the unmodified natural duplex ($T_m$ = 67 °C). It should be noted that the *Tm* value for the duplex **ODN7/DNA1** (**Y** = dG) or **RNA1** (**Y** = rG) was 54 °C and 57 °C respectively, higher than the other duplexes. Taken together, our results of the $T_m$ measurement suggest that AOVT might form stable triple hydrogen bonds with guanine to orient the C6-vinyl group of AOVT toward the opposite side of the complementary bases. Next, we attempted the crosslinking reactions to the nucleic acids-binding protein using the duplex DNA bearing AOVT derivative **2**.

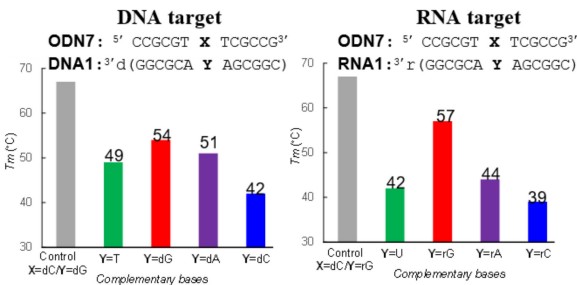

**Figure 5.** Comparative thermal denaturing analysis of duplexes **ODN7/DNA1** or **RNA1**. $T_m$ values were measured in 50 mM MES buffer (pH 7.0) containing 100 mM NaCl.

### 3.3. Crosslinking Reactions to DNA Methyl Transferase

We chose cytosine-5-DNA methyl transferase (DNMT) as a model DNA binding protein. DNMT performs the methylation of cytosines for epigenetic modification of the genome that is involved in regulating many cellular processes [6]. In addition, aberrant methylation can cause many diseases, such as cancer [22–24]. The efficient inhibition of DNMT offers the possibility of interfering with the methylation process, which may allow control of the epigenetic regulation of cells and treat various cancers [25].

The mechanism of the cytosine-5 methylation with DNMT is illustrated in Figure 6A. A thiol of the Cys residue in the active site of DNMT acts as a nucleophile and attacks the C6 position of the cytosine to form an enzyme-linked intermediate. The resulting nucleophilic C5 position of the cytosine is then methylated by S-adenosyl-l-methionine (SAM). Subsequent abstraction of the proton at the 5 position and b-elimination provides the 5-substituted pyrimidine and active enzyme (Figure 6A) [26]. Based on this mechanism, the modified bases forming a covalent bond to DNMT were reported as irreversible inhibitors [27–31]. We expected that *acy*AOVT (**2**) replacement of the cytosine methylated by DNMT can react with the Cys residue in this enzyme (Figure 6B). In our investigation, we used the bacterial DNMT, i.e., the commercially-available Hha1 DNMT-1 from *Haemophilus haemolyticus*. Hha1 DNMT-1 methylated the internal 2′-deoxycytidine in the target sequence 5′-GCGC-3′, in which AOVT was substituted in place of the internal 2′-dC. We synthesized **ODN6**, which contained AOVT in the Hha1 DNMT-1 recognition sequence and evaluated the crosslink reactivity to DNA. The base selectivity was similar, but the reactivity to the target DNA was significantly lower compared to that of the previous sequence (Figure S12).

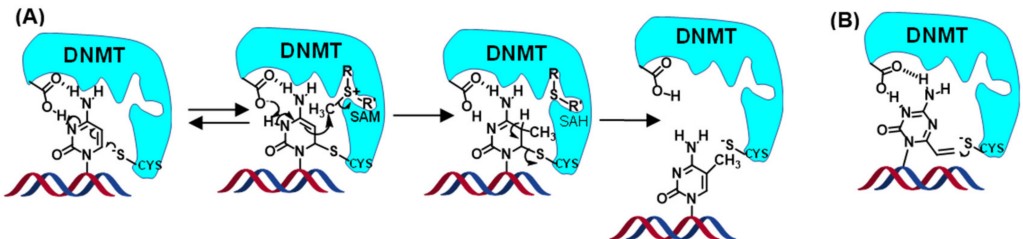

**Figure 6.** (**A**) Plausible mechanism for methylation by cytosine-5 methyltransferases (DNMTs) with SAM as a cofactor (**B**) Possible mechanism for covalent trapping with AOVT (**2**) derivative.

We selected DNA2 containing methyl cytosine as a target DNA because Dnmt1 methylates hemimethylated CpG sites on one strand of double-stranded DNA. The methylation conditions were determined using the duplex DNA between DNA**2** and DNA**3** contained cytosine instead of acyclic AOVT and the mono-methylated product in DNA**3** was confirmed by MALDI-TOF MS. The duplex DNA (**ODN6** and **DNA2**) was incubated with Hha1 DNMT-1 in the reaction buffer containing 5 mM 2-mercaptoethanol at 37 °C in the determined conditions. The reaction mixture was analyzed by sodium dodecyl sulfate −15% polyacrylamide gel electrophoresis (SDS−PAGE) and visualized with

silver staining. The slower mobility band (*) was observed around 51kD and not observed with the non-reactive duplex DNA and Hha1 DNMT-1 (Figure 7A). We performed the reaction using the 5′ FAM-labelled **ODN6** and analyzed by SDS-PAGE. The slower mobility band stained silver matched the fluorescent band in the reaction using the 5′ FAM-labelled **ODN6** (Figure 7B) and this band was derived from the complex between Hha1 DNMT-1 and the duplex DNA. The quantification of the products was next performed by using the Flamingo^TM Gel staining in-gel fluorescence analysis. We confirmed the linear correlation between the protein amount and the fluorescence intensity. Thus, the complex between the protein and ODN can be quantified by measuring the fluorescence intensity on the gel (Figure 7C). The reactions were performed at 37 °C and 4 °C. The yields were obtained by quantification of the fluorescence intensity for each band and reached 77% at 37 °C after 24 h.

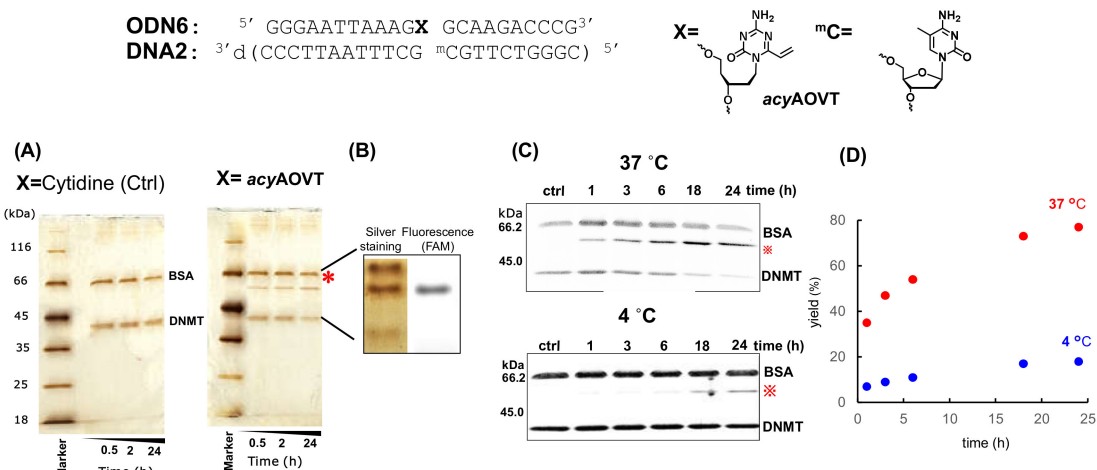

**Figure 7.** Evaluation of the crosslinking reaction between duplex DNA and Hha1 DNMT-1 (**A**) SDS-Page gel stained silver for analysis of reactions. Duplex DNA (**ODN6** (**X** = C or *acy*AOVT) and **DNA2**: 3 μM) and Hha1 DNMT-1 (1 μM) was incubated in the presence of SAM at 4 °C. (**B**) 5′-FAM labelled **ODN6** was used in the same reaction of (**A**) and the gel was visualized by silver staining and fluorescence. (**C**) SDS-Page gel stained Flamingo^TM for analysis of reactions. Duplex DNA (**ODN6** (**X** = C for control or *acy*AOVT) and **DNA2**: 1 μM) and Hha1 DNMT-1 (1 μM) was incubated in the presence of SAM at 4 °C or 37 °C. (**D**) Time course of the yields at 4 °C or 37 °C.

The yields significantly decreased at 4 °C (Figure 7D), suggesting that the reaction rate depends on the reaction temperature. The reactions were performed using ODN**6** and DNA**4** containing 4 kinds of bases (G, A, C and T) at the complementary site for *acy*AOVT. The crosslinking yields to these targets were similar to that observed with G (Figure S13). These results suggested that DNMT can bind to the mismatch sequences and make a transition to flip out of the target *acy*AOVT to induce the crosslinking reactions. Taken together, the results of the evaluation for the reaction to Hha1 DNMT-1 indicated that the *acy*AOVT in the duplex would react with Hha1 DNMT-1 even in the presence of 2-mercaptoethanol.

## 4. Discussion

We have synthesized ODN containing the *acy*AOVT base **2** that is expected to react with the nucleophilic residue of the nucleic acid-binding protein in the proximity of the vinyl group. The reactivity with *acy*AOVT to DNA or RNA was significantly higher than that of *acy*AOVP. The interesting result that only one nitrogen substitution on AOVP drastically increased the reactivity could be attributed to the electron withdrawing effect of the nitrogen atom. The *acy*AOVT (**2**) showed relatively high yields to dC, dT and dG in DNA and rC, rU in RNA. The highest reactivity to cytosine with the AOVT derivatives **1** and **2** might be due to the high nucleophilicity of the amino group in cytosine and to the flexible linker, which makes it possible to access the vinyl group in the amino group of the cytosine. The cross-linking yields to adenine in the DNA or RNA with *acy*AOVT **2** were

significantly lower than that of Et-AOVT **1**. The vinyl group in *acy*AOVT **2** might be located away from the reactive site of adenine because of being positioned at similar natural base-pair by the short linker with **2**, resulting in the low crosslinking yields to adenine (Figure S14). The melting temperature ($T_m$) values of the duplex for **ODN7/DNA1(G)** or **RNA1(G)** were the highest in all the duplexes containing the other AOVT-base pairing. These results suggested that the duplex containing *acy*AOVT-G would be stabilized by forming three hydrogen bonding, resulting in orientation of the vinyl group in *acy*AOVT to the opposite side of the complementary bases (Figure 8).

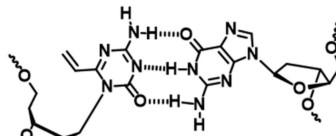

**Figure 8.** Hypothetical structure for *acy*AOVT and guanine in duplex DNA.

We attempted the reaction to the commercially-available DNMT using the duplex containing *acy*AOVT-G as a model reaction with the nucleic acids-binding protein. The results indicated that *acy*AOVT might react with the thiol group of the cysteine residue in DNMT in close proximity as shown in Figure 9.

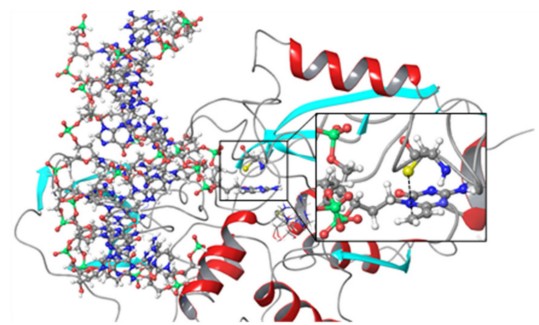

**Figure 9.** Predictive structure for reaction intermediate between *acy*AOVT and cysteine in *DNMT* (PDB-5MHT).

We expect that this *acy*AOVT-G base pairing system would provide a "cross-linkable duplex material" to react with the nucleic acid-binding proteins.

## 5. Conclusions

We have designed *acy*AOVT **2** as a reactive base to the nucleic acid-binding protein. The ODN containing **2** exhibited a higher reactivity to the pyridine bases (C, T and U) at the complementary site of **2** in duplexes DNA-DNA and DNA-RNA than that of *acy*AOVP. The results of the $T_m$ measurements suggested that *acy*AOVT **2** can form a stable base pair with G, and consequently, the vinyl group of **2** might be located on the opposite side of the complementary bases. The preliminary results for the reaction to DNMT with the duplex DNA containing *acy*AOVT demonstrated the potential of *acy*AOVT-G base pairing system in the reaction to the nucleic acids-binding protein. This system can be used for the crosslinking reaction to the RNA binding protein. We expect that our system will provide a useful tool for the molecular study of the RNA binding protein in control of the RNA biological functions.

**Supplementary Materials:** The following are available online at http://www.mdpi.com/2076-3417/10/21/7709/s1.

**Author Contributions:** K.O. and K.Y. conceived and designed the experiments. K.O. and S.I. performed the experiments. F.N. and H.O. wrote the study. F.N. supervised the project. All authors have read and agreed to the published version of the manuscript.

**Funding:** This study was supported by a Grant-in-Aid for Scientific Research on Innovative Areas "Middle Molecular Strategy" (No. JP15H05838) from the Japan Society for the Promotion of Science (JSPS). This work was supported in part by "Dynamic Alliance for Open Innovation Bridging Human, Environment and Materials" from the Ministry of Education, Culture, Sports, Science and Technology of Japan (MEXT).

**Conflicts of Interest:** The authors declare no conflict of interest.

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
