# Peer review of "Design of the Crosslinking Reactions for Nucleic Acids-Binding Protein and Evaluation of the Reactivity"

_applsci, doi:10.3390/app10217709_

Round 1
Reviewer 1 Report
This paper described crosslink reagent for nucleic acid. This paper didn’t include any important points, and the authors should address the following comments:
- Figure 1: (A) "R:" overlaps the figure. Please correct the figure.
- The caption of Sheme 1 does not match the experimental term.
Lines 115: 58%--> sheme1 (a) 64%
Lines 130: 55 °C, 23 h--> sheme1 (b) 50 °C, 24 h
Lines 134: 7h --> sheme1 (c) 2h
Lines 138: 51% --> sheme1 (c) 54%
Lines 148: 19h --> sheme1 (d) 1h
Lines 165: 52% --> sheme1 (e) 54%
Lines 191: 1h --> 45 min
- Lines 159 2.2.3 --> 2.2.4
- Lines 172 2.2.4 --> 2.2.5
- Lines 187 2.2.5 --> 2.2.6
- Lines 179: to give 27 --> to give 9
- Lines 210: K2CO3/MeOH à 45 mM K2CO3/MeOH containing 10 mM 1-Octanethiol ?
- Lines 217-219 does not match caption of Scheme 2 (c)-(d).
- Lines 219: ODN 3 -->ODN 5 or ODN 6?
- In Figure 3 does not descrived “A””B” or “C”. Authors shows Figures 3A, B and C in lines 296 and 313. Please show A-C in Figure 3, and include a description of A-C in the caption.
- Figure 4: Tha caption in not enouph. Please include a description of A-C in the caption. The explanation of Y=dC, T, dG, dA in the graph is insufficient. The "RNA" sign at the top of the RNA graph has disappeared.
- Figure 5: DNA target, Tm value of dA is disappeared.
Author Response
Reviewer: 1
[Comment 1] Figure 1: (A) "R:" overlaps the figure. Please correct the figure.
[Answer] We have corrected Figure 1
[Comment 2] The caption of Scheme 1 does not match the experimental term.
[Answer] We have changed the caption of scheme 1 matched to the experimental sections..
[Comment 3] Lines 159 2.2.3 --> 2.2.4; Lines 172 2.2.4 --> 2.2.5; Lines 187 2.2.5 --> 2.2.6; Lines 179: to give 27 --> to give 9; Lines 210: K2CO3/MeOH a 45 mM K2CO3/MeOH containing 10 mM 1-Octanethiol ?; Lines 217-219 does not match caption of Scheme 2 (c)-(d).; Lines 219: ODN 3 -->ODN 5 or ODN 6?
[Answer] We have corrected all suggested points.
[Comment 4] In Figure 3 does not descrived “A””B” or “C”. Authors shows Figures 3A, B and C in lines 296 and 313. Please show A-C in Figure 3, and include a description of A-C in the caption.
[Answer] We are very sorry for our errors. We mistook the Figure 4A~C in line 296 and 313.
[Comment 5] Figure 4: The caption in not enough. Please include a description of A-C in the caption. The explanation of Y=dC, T, dG, dA in the graph is insufficient. The "RNA" sign at the top of the RNA graph has disappeared.
[Answer] We have added the description of Figure 4 and corrected of Figure 4.
[Comment 6] Figure 5: DNA target, Tm value of dA is disappeared.
[Answer] We have corrected Figure 5.
Reviewer 2 Report
The manuscript entitled “Design of the crosslinking reactions for nucleic acids binding protein and evaluation of the reactivity” by Nagatsugi and co-workers reported the highly reactive electrophile containing cytosine analog of AVOT, its ODN with acyclic nucleotide modification, and its reactivity towards both the complementary sequence and DNMT. The concept of this work is original, and the application shown in this paper is fascinating for the nucleic acid chemistry field. Thus, I strongly recommend the paper will be published after minor alterations.
Major concerns:
- In figure 7, the crosslinking was conducted with DNA2 where AOVT in ODN6 can make base pairing with G so that the electrophile moiety faces outside of duplex where DNMT contacts to the ODN. This is a convincing design for POC but the control experiment is missing. The author should compare the other nucleobase oligonucleotides where the vinyl group of AOVT can face inside of duplex (cf. figure 4).
- The author claimed AOVT is superior to AOVP in terms of the reactivity towards C or T (U), but it can potentially cause non-specific Michael addition to the vinyl moiety. To address this issue, the crosslinking experiment in figure 7 should be carried out in the presence of reduced glutathione or DTT.
Minor points;
- In line 49, “chloacetamide” looks typo.
- In figure 1B, to compare the natural or AOVT positions clearer, coloring should be changed (for example, natural in magenta, AOVT in green)
- A table for ODNs, DNAs, and RNAs should help to understand.
- Figure 1S and 2S (mentioned in line 355 and 393) are missing but look quite important.
- Why did author choose DNA2 containing methylcytosine, which is the product of DNMT but not the substrate in figure 7?
- In line 385, “nitrogen substitution” sounds proper rather than “nitrogen addition”.
- In figure S1–S11, the figure captions are missing.
- The MALDI-MS results of crosslinked adduct (for example, ODN5 and DNA1) is missing. MS results will make this work convincing.
- In figure 5, the yield of Y=dA in the DNA target is invisible.
- Figure S12 was not mentioned in the main text. This observation is interesting to compare with Figure 4.
Author Response
Reviewer: 2
Major concerns:
[Comment 1] In figure 7, the crosslinking was conducted with DNA2 where AOVT in ODN6 can make base pairing with G so that the electrophile moiety faces outside of duplex where DNMT contacts to the ODN. This is a convincing design for POC but the control experiment is missing. The author should compare the other nucleobase oligonucleotides where the vinyl group of AOVT can face inside of duplex (cf. figure 4).
[Answer] We have added the experimental results using the other nucleobases in supporting information (Figure S13). In addition, the comments on these experimental results as below.
The reactions were performed using ODN6 and DNA4 containing 4 kinds of other bases (G, A, C and T) at the complementary site for acyAOVT. The crosslinking yields to these targets were similar to that observed with G (Figure S13). These results suggested that DNMT can bind to the mismatch sequences and make a transition to flip out of the target acyAOVT to induce the crosslinking reactions.
[Comment 2] The author claimed AOVT is superior to AOVP in terms of the reactivity towards C or T (U), but it can potentially cause non-specific Michael addition to the vinyl moiety. To address this issue, the crosslinking experiment in figure 7 should be carried out in the presence of reduced glutathione or DTT.
[Answer] The crosslinking experiments in Figure 7 were carried out in the presence of 5 mM 2-mercaptoethanol, which has a free thiol group. In these conditions, the Michael addition with thiol group to vinyl moiety might not occur.
Minor points
[Comment 1] In line 49, “chloacetamide” looks typo.
[Answer] We have corrected the spell.
[Comment 2] In figure 1B, to compare the natural or AOVT positions clearer, coloring should be changed (for example, natural in magenta, AOVT in green)
[Answer] We have changed the color in Figure 1B and C to follow the reviewer’ s suggestion.
[Comment 3] A table for ODNs, DNAs, and RNAs should help to understand.
[Answer] We have added the Table for the sequences ODNs, DNAs and RNAs in supporting information.
[Comment 4] Figure 1S and 2S (mentioned in line 355 and 393) are missing but look quite important.
[Answer] We are sorry for our errors. In line 355 and 393, Figure S12 and Figure S14 are correct and we have corrected these errors.
[Comment 5]. Why did author choose DNA2 containing methylcytosine, which is the product of DNMT but not the substrate in figure 7?
[Answer] We have described the comments on the DNA target containing as below.
We selected DNA2 containing methyl cytosine as a target DNA because Dnmt1 methylates hemimethylated CpG sites on one strand of double-stranded DNA. The methylation conditions were determined using the duplex DNA between DNA2 and DNA3 contained cytosine instead of acyclic AOVT and the methylated product in DNA3 was confirmed by MALDI-TOF MS. The duplex DNA (ODN6 and DNA2) was incubated with Hha1 DNMT-1 in the reaction buffer containing 5 mM 2-mercaptoethanol at 37 ˚C in the determined conditions.
[Comment 6] In line 385, “nitrogen substitution” sounds proper rather than “nitrogen addition”.
[Answer] We have changed to “nitrogen substitution“.
[Comment 7] In figure S1–S11, the figure captions are missing.
[Answer] We have added the Figure captions for Figure S1-S11.
[Comment 8] The MALDI-MS results of crosslinked adduct (for example, ODN5 and DNA1) is missing. MS results will make this work convincing.
[Answer] We did not measure the MALDI-MS for crosslinked adducts, because we have reported the crosslinked adducts using the similar derivative (1)
[Comment 9] In figure 5, the yield of Y=dA in the DNA target is invisible.
[Answer] We have corrected Figure 5.
[Comment 10] Figure S12 was not mentioned in the main text. This observation is interesting to compare with Figure 4
[Answer] We are sorry for our errors. We have mistaken Figure S12 for Figure 1S and corrected this error.